# Self-Organizing Visual Prototypes for Non-Parametric Representation Learning

**Thalles Silva** [1]   **Helio Pedrini** [1]   **Adín Ramírez Rivera** [2]

## Abstract

We present Self-Organizing Visual Prototypes (SOP), a new training technique for unsupervised visual feature learning. Unlike existing prototypical self-supervised learning (SSL) methods that rely on a single prototype to encode all relevant features of a hidden cluster in the data, we propose the SOP strategy. In this strategy, a prototype is represented by many semantically similar representations, or support embeddings (SEs), each containing a complementary set of features that together better characterize their region in space and maximize training performance. We reaffirm the feasibility of non-parametric SSL by introducing novel non-parametric adaptations of two loss functions that implement the SOP strategy. Notably, we introduce the SOP Masked Image Modeling (SOP-MIM) task, where masked representations are reconstructed from the perspective of multiple non-parametric local SEs. We comprehensively evaluate the representations learned using the SOP strategy on a range of benchmarks, including retrieval, linear evaluation, fine-tuning, and object detection. Our pre-trained encoders achieve state-of-the-art performance on many retrieval benchmarks and demonstrate increasing performance gains with more complex encoders. Code: https://github.com/sthalles/sop.

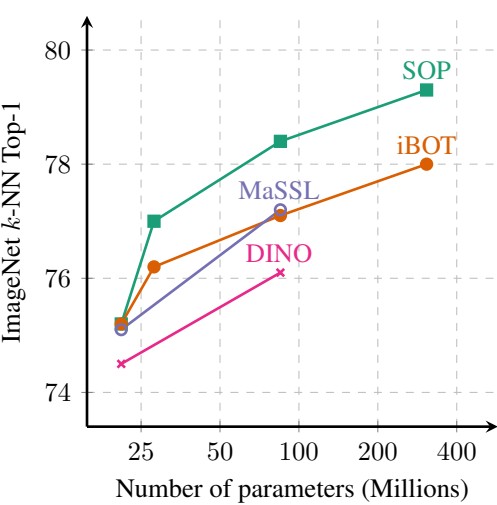

*Figure 1.* $k$-**NN top-1 accuracy on ImageNet.**

## 1. Introduction

State-of-the-art self-supervised learning (SSL) methods (Zhou et al., 2022; Oquab et al., 2023) for computer vision rely on *prototype* learning with a *multiview* training strategy using *joint-embedding architectures* (JEA). These methods have a similar framework: they learn a *large set* of pre-defined prototypes that are presumed to represent hidden clusters in the data. When presented with multiple views as input, these methods assume that prototypes must produce consistent predictions across views of the same image during pre-training. To avoid ill-posed solutions, methods enforce an equipartition constraint (Asano et al., 2019) that uniformly assigns a batch of images to prototypes using techniques such as centering (Caron et al., 2020) and Sinkhorn-Knopp (Cuturi, 2013). If these regularizers are removed, pre-training collapses to trivial solutions.

Despite undeniable success, this framework places excessive importance on the prototypes' representations, which need to encode a comprehensive set of relevant features for consistent and accurate predictions across views. Current methods (Caron et al., 2020; Zhou et al., 2022; Caron et al., 2021; Oquab et al., 2023) report their best performance when the number of prototypes $K \gg C$ is much larger than the number of actual classes $C$ in the data. However, in addition to increased computational and memory requirements, we argue that over-clustering is a suboptimal approach to fully exploring the feature space under the weak supervision of SSL pre-training. Specifically, a large $K$ translates into fewer images per prototype, facilitating the optimization problem by biasing simpler features, which may limit the complexity of the learned features.

---

[1]Institute of Computing, University of Campinas, Campinas-SP, Brazil [2]Department of Informatics, University of Oslo, Oslo, Norway. Correspondence to: Thalles Silva <thalles.silva@students.ic.unicamp.br>, Helio Pedrini <helio@ic.unicamp.br>, Adín Ramírez Rivera <adinr@uio.no>.

*Proceedings of the $42^{nd}$ International Conference on Machine Learning*, Vancouver, Canada. PMLR 267, 2025. Copyright 2025 by the author(s).

We hypothesize that when a large number of prototypes is employed, they may fail to capture the essential features needed to fully describe the hidden clusters within the data. Each cluster can be conceptualized as a "region" of salient features in space. If a prototype underrepresents its underlying region, the interactions with views may not be strong enough to pull representations towards it. Furthermore, over-clustering does not ensure comprehensive coverage of the feature space. Even with a uniform distribution on the hypersphere (Wang & Isola, 2020), gaps may still exist between prototypes, resulting in features that cannot be effectively attracted to any prototype due to insufficient supervision.

Motivated by these limitations, we deviate from the popular prototypical learning paradigm and propose a non-parametric approach that optimizes the feature space using Self-Organizing Prototypes (SOP). Our proposal organizes the feature space by exploiting *random* local structures within the data. Intuitively, an SOP can be viewed as a data structure, such as a graph, representing a local region in the feature space. Each SOP consists of a set of non-parametric *support embeddings* (SEs), which reside in close proximity and share semantic characteristics. Each support embedding predicts the degree of similarity (or likelihood) of a view to its "region" in latent space (SOP), based on its distinct feature set. Then, SEs combine their individual votes to produce the final likelihood of a view to their SOP. Unlike prototypical SSL, SOPs are inherently stochastic. Instead of optimizing a set of prototypes that move in space during pre-training, SOPs randomly select and optimize regions in the data.

Unlike regular methods (Caron et al., 2020; Zhou et al., 2022), SOPs are dynamic (not constrained to a region in space) and fully cover the space of features through built-in randomization. SOPs explore their surroundings to bootstrap SEs, which in turn augment the feature set used to represent their region in space. By comparison, SOPs can be viewed as dynamic, feature-enriched versions of regular prototypes. While prototypical SSL optimizes relationships between views and prototypes (discrete points in space), our method optimizes views over SOPs, i.e., local "regions" in space containing multiple SEs that cover a larger area of the same region, offering better adaptation. We demonstrate that our dynamic SOP structure alleviates the over-clustering problem inherent in prototypical SSL (cf. Table 10). To ensure that our learned representations contain information beyond the class level, we adapt the Masked Image Modeling (MIM) pretext task using the SOP strategy and introduce a novel pretext task where masked representations are trained to agree with corresponding non-masked ones from the perspective of multiple local-level supports.

Our contributions are threefold:

- We present Self-Organizing Prototypes, an alternative approach to prototypical SSL. We optimize views based on the soft similarity perspective of Self-Organizing Prototypes in the space of non-parametric representations.
- We propose the novel SOP-MIM pretext task to learn fine-grained features by reconstructing local-level masked representations based on non-parametric SOPs composed of patch-level feature as local supports. We show that SOP-MIM improves downstream performance on dense prediction tasks compared to existing solutions.
- Our work reaffirms the feasibility of non-parametric SSL, where we avoid learning prototypes from random weights. We show that our method is stable, does not require extra regularizers to avoid mode collapse, is extensible to many pretext tasks such as MIM, does not require over-clustering, cf. Table 10, and produces transferable representations. Moreover, we show that SOP's performance improves as the model architecture is scaled.

## 2. Methodology

We argue that the common approach of learning prototypes as bottleneck feature vectors to represent salient regions in feature space is suboptimal due to the lack of strong supervision and the inductive biases that promote over-clustering in SSL. The lack of a strong supervisory signal promotes under-optimized regions in the latent space, which are typically modeled as clusters represented by prototypes. Regular prototypical SSL contrasts different views against these prototypes using point-to-point comparisons, resulting in unstable comparisons during the learning process. Instead, we propose optimizing views over local structures called Self-Organizing Prototypes (SOPs). We define an SOP as a set consisting of an anchor prototype and its $k$ closest neighbors, which we call support embeddings (SEs). Unlike prototypical SSL, SOPs leverage local structures within the data and optimize regions in the feature space (via SEs) rather than individual points, as sources of representative features. SOPs can be viewed as augmented prototypes. However, rather than relying on a single prototype, SOPs group features from different SEs, which collectively represent the region in space more accurately, cf. Fig. 2.

**Notation.** Let $X$ be an image dataset and $x \sim X$ a uniformly random observation. We denote by $x^v$ the $v$-th augmented version of $x$, referred to as a view of $x$, where the superscript $v$ indexes the views $V$. To create views, we use a random transformation function $t$ such that $x^v = t(x)$. For simplicity, we consider the case where $V = 2$. However, we explore multiple view scenarios in the main experiments. We denote by $f_\Phi$ a Vision Transformer (ViT) (Dosovitskiy et al., 2020) encoder with parameters $\Phi$ that receives a view and produces a matrix of representation vectors $Z^v = f_\Phi(x^v) \in \mathbb{R}^{L \times d}$, where $L$ and $d$ are the number

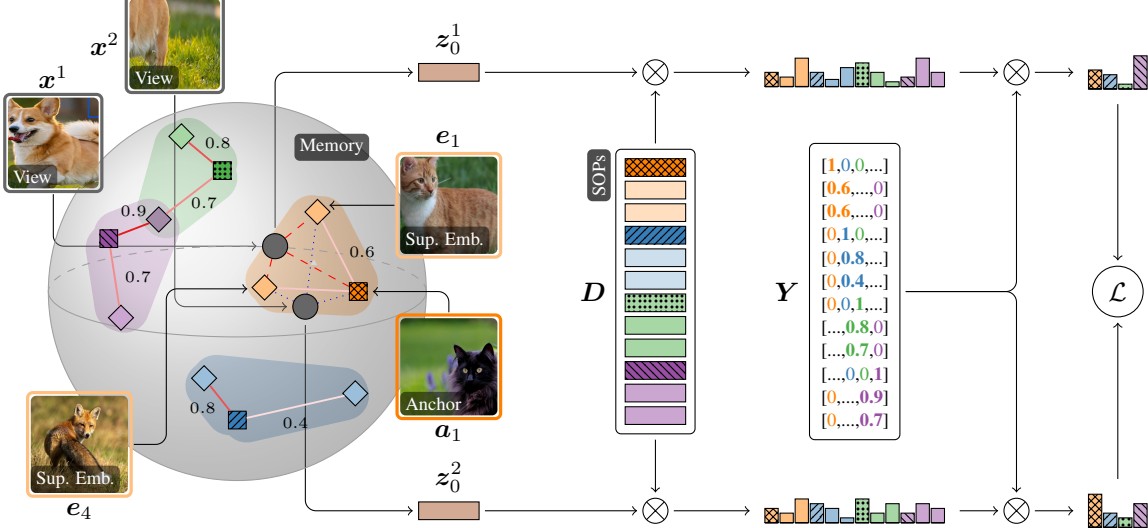

*Figure 2.* First, we select a set of random anchors $\boldsymbol{A} = \{\boldsymbol{a}_i\}_{i=0}^{K}$ (colored squares with patterns) from a set of representations kept in memory (gray sphere). Second, each anchor selects $k$ support embeddings (SEs) (colored diamonds) as their nearest neighbors (2 in this illustration). Each anchor $\boldsymbol{a}_i$ and their SEs form an SOP, representing a hidden structure within the data (shaded colored region). Note that a given embedding may belong to more than one SOP simultaneously. Put together, SOPs can be linearly arranged as a dataset $\boldsymbol{D} \in \mathbb{R}^{K(k+1) \times d}$ with labels $\boldsymbol{Y} \in \mathbb{R}^{K(k+1) \times K}$ representing the interconnections between SEs and anchors. Intuitively, each SOP contains a set of SEs that estimate the degree of similarity between views and SOPs. Then, SEs combine their votes to produce a final score for each view, resulting in similarity distributions optimized to be consistent across SOPs.

of patch tokens and feature dimensionality respectively, such that $\boldsymbol{Z}^v = \{\boldsymbol{z}_l\}_{l=0}^{L}$ contains patch representations where the first element $\boldsymbol{z}_0 \in \mathbb{R}^d$ is the classification or [CLS] token embedding and the remaining $\boldsymbol{Z}_{1:L,:}$ elements are patch embeddings from an image $\boldsymbol{x}$. Instead of learning class- and patch-level discrete features (or prototypes) as previous work did (Caron et al., 2020; Zhou et al., 2022), we define memory containers $\boldsymbol{E}_C \in \mathbb{R}^{N_C \times d}$ and $\boldsymbol{E}_P \in \mathbb{R}^{N_P \times d}$ to store [CLS] and patch embeddings from previous iterations. Intuitively, the memories act as subsets of the training data features for global and local representations. For details regarding practical experiments, cf. Appendix A.

Next, we introduce the basic functionality of current prototypical SSL methods and highlight the two commonly used loss functions. Then, we present SOP, emphasize its main differences and advantages.

### 2.1. Learning Representations with Prototypical SSL

Current prototypical SSL methods (Oquab et al., 2023; Zhou et al., 2022) have a nearly identical framework composed of two pretext tasks: (i) cluster assignment prediction over class-level embeddings and (ii) token-level embedding reconstruction or Masked Image Modeling (MIM). Usually, each task is learned with a different set of trainable parameters.

The cluster assignment prediction task aims to learn embeddings that covary w.r.t. a set of *learnable* prototypes $\theta \in \mathbb{R}^{K \times d}$. The optimization follows

$$\mathcal{L}_{\text{[CLS]},\theta} = - \sum_{\boldsymbol{x} \sim \boldsymbol{X}} P_\theta^{\text{[CLS]}}(\boldsymbol{z}_0^1)^T \log\left(P_\theta^{\text{[CLS]}}(\boldsymbol{z}_0^2)\right), \quad (1)$$

where $P_\theta^{\text{[CLS]}}(\boldsymbol{u}) = \sigma(\langle u, \theta^T \rangle)$, $\sigma(\cdot)$ is the softmax function and $\langle \cdot, \cdot \rangle$ is the cosine similarity. Basically, $P_\theta^{\text{[CLS]}}(\cdot)$ is a linear layer, parameterized by $\theta$, that maps the views' vector embeddings $\boldsymbol{z}^v$ into $K$ pseudo-categories assigning each representation a *soft* distribution (prediction) that describe its membership probabilities to all prototypes.

This objective can be viewed from a pseudo-clustering perspective, where each prototype represents a pseudo-class. In both parametric and non-parametric approaches, each prototype estimates the likelihood of a view to itself. Existing solutions typically employ a single prototype, either learnable (Caron et al., 2021) or non-parametric (Silva et al., 2024), to represent a region in the representation space.

### 2.2. Self-Organizing Prototypes

The under-representation problem in prototypical SSL occurs when a prototype lacks the necessary features to describe its region in space. We propose to solve this problem by increasing the information redundancy of a prototype by

augmenting its feature set through Self-Organizing Prototypes (SOP).

SOPs explore random local structures in the latent space containing different embeddings that share semantic characteristics about their region. These local structures are used to bootstrap support embeddings (SEs), leveraged as a source of representative features. This approach assumes that embeddings in a vicinity contain enough information to describe their region in the feature space. Ideally, each SE within an SOP contributes complementary features that may not be present or sufficiently emphasized in a single prototype.

### 2.2.1. NON-PARAMETRIC SUPPORT EMBEDDING SELECTION

An important consideration is how to bootstrap SEs while maintaining semantics. If SEs do not share semantic characteristics, their contributions may be noisy, potentially harming the learned features. Inspired by recent work on non-parametric SSL (Silva et al., 2024), we use a memory $\boldsymbol{E}_\mathrm{C}$ to store representations from previously processed images during training. From the memory $\boldsymbol{E}_\mathrm{C}$, we perform three main operations: *store*, *sample*, and *search*.

To build an SOP, first, we sample a subset of anchor representations $\boldsymbol{A} = \{\boldsymbol{a}_i\}_{i=0}^K \subset \boldsymbol{E}_\mathrm{C}$ from memory. Second, we search for SEs through spherical $k$-Nearest Neighbors with anchors $\boldsymbol{A}$ as centroids such that $\boldsymbol{D} = \arg\max_e^k \left(\langle \boldsymbol{A}, \boldsymbol{E}_\mathrm{C}^T \rangle\right)$, where the $\arg\max_e^k$ operator returns the anchors and the set of the top-$k$ closest neighbors of each anchor. At this point, $\boldsymbol{D}$ can be viewed as a dataset containing $K$ SOPs, each containing $k+1$ members, i.e., an anchor $\boldsymbol{a}_i$, and $k$ SEs. Note that this definition allows an SE $\boldsymbol{e}_j \in \boldsymbol{E}$ to belong to more than one SOP, cf. Fig. 2.

Given the potential uncertainty in the unsupervised $k$-NN selection algorithm, it is reasonable to expect that SEs (within an SOP) contribute differently. Naturally, SEs that share more features with an anchor should have a stronger influence than those more distinct. Based on that observation, we introduce $\boldsymbol{Y}$ to model the contributions of SEs within their SOPs. Intuitively, we can think of an SOP as an acyclic directed graph (DAG) containing $k+1$ nodes (an anchor plus SEs) and $k-1$ edges. All edges point to the *sink* anchor node, and the strength of the edges represents the contributions of SEs towards its SOP, cf. Fig. 2. A naive strategy would assign equal contribution to each SE, i.e., $\boldsymbol{Y}$ expressed as a one-hot vector. We show in Section 4 that such a strategy is suboptimal, likely due to false positives from the $k$-NN selection. Instead, we propose $\boldsymbol{y} \in \boldsymbol{Y}$, to model the *soft contribution* of an SE as the embedding similarity score with anchors, i.e., $\langle \boldsymbol{e}_j, \boldsymbol{a}_i \rangle$ for $j \in \{0, 1, \ldots, K(k+1)\}$ and $i \in \{0, 1, \ldots, K\}$. This strategy minimizes potential noise arising from false positives from the SE selection algorithm.

Now, we can compute the probability distributions for each view as $P^{[\mathrm{CLS}]}(\boldsymbol{u}) = \sigma(\langle \boldsymbol{u}, \boldsymbol{D}^T \rangle)\boldsymbol{Y}$, where $\boldsymbol{Y}$ are soft contributions that sum up to one and encode the contributions of each SE. Compared to the parametric loss (1), our approach swaps the learnable prototypes $\theta$ for non-parametric SOPs. Note that the matrix multiplication between the probability distribution $\sigma(\cdot)$ and soft contributions $\boldsymbol{Y}$, represents the weighted combination of the SEs predictions within each SOP.

Finally, we minimize the non-parametric version of $\mathcal{L}_{[\mathrm{CLS}],\theta}$ (1), as

$$\mathcal{L}_{[\mathrm{CLS}]} = -\sum_{\boldsymbol{x} \sim \boldsymbol{X}} P^{[\mathrm{CLS}]}(\boldsymbol{z}_0^1)^T \log\left(P^{[\mathrm{CLS}]}(\boldsymbol{z}_0^2)\right). \quad (2)$$

### 2.3. Self-Organizing Prototypes for Masked Image Modeling

The MIM task has been extensively explored by Zhou et al. (2022) and Oquab et al. (2023) from the parametric perspective. The task aims to produce consistent predictions between reconstructed patch embeddings and their corresponding uncorrupted representations w.r.t. a set of learnable discrete local-level features. The goal is to train an *online* local-level tokenizer $\phi$ by randomly masking a portion of the patch token representations $\boldsymbol{x} = \{\boldsymbol{x}_l\}_{l=0}^L$ using a binary mask $\boldsymbol{m} \in \{0,1\}^L$ such that $\hat{\boldsymbol{x}} = \{\hat{\boldsymbol{x}}_i : (1 - m_i)\boldsymbol{x}_i + m_i\boldsymbol{e}_{[\mathrm{MASK}]}\}^L$ is a corrupted version of the input image $\boldsymbol{x}$, and $\boldsymbol{e}_{\mathrm{MASK}}$ is a learnable token. The corrupted input $\hat{\boldsymbol{x}}$ is fed to the encoder $\hat{\boldsymbol{Z}} = f(\hat{\boldsymbol{x}})$ and reconstructed from the uncorrupted version following

$$\mathcal{L}_{\mathrm{patch},\phi} = -\sum_{l=1}^L m_l P_\phi^{\mathrm{patch}}(\boldsymbol{z}_l^1)^T \log\left(P_\phi^{\mathrm{patch}}(\hat{\boldsymbol{z}}_l^1)\right), \quad (3)$$

where, similar to $\mathcal{L}_{[\mathrm{CLS}],\theta}$ (1), $P_\phi^{\mathrm{patch}}(\cdot)$ is a linear layer that computes the probability distributions w.r.t. learnable discrete features $\phi$ by soft assigning the patch tokens to $\dot{K}$ distinct discretized representations. Note that the loss $\mathcal{L}_{\mathrm{patch},\phi}$ (3) skips the $[\mathrm{CLS}]$ token $\boldsymbol{x}_0$, and optimizes different versions of the *same* image view, where one is masked.

We propose a new version of the MIM pretext task based on a non-parametric strategy. Instead of learning a set of discrete features (online tokenizer), we obtain the probability distributions $P^{\mathrm{patch}}(\cdot)$ by exploring relationships between semantically similar patch embeddings in the space of non-parametric representations using Self-Organizing Prototypes, cf. Section 2.2.

We start by randomly sampling $\dot{K}$ anchor patch discrete tokens $\dot{\boldsymbol{A}} = \{\dot{\boldsymbol{a}}_j\}_{j=0}^{\dot{K}} \subset \boldsymbol{E}_\mathrm{P}$ as the roots for our SOPs. Then, each anchor selects $k$ nearest patch token representations to become SEs. In conjunction, anchors and SEs form patch-level SOPs, and $\dot{\boldsymbol{D}} = \arg\max_e^k \left(\langle \dot{\boldsymbol{A}}, \boldsymbol{E}_\mathrm{P}^T \rangle\right)$ is the dataset

containing a set of SOPs. Note that $\boldsymbol{E}_{\mathrm{P}}$ can be seen as a non-parametric or *offline* tokenizer.

Similarly to Section 2.2, we obtain the patch-level probability distributions, in a non-parametric form, as $P^{\mathrm{patch}}(\boldsymbol{v}) = \sigma\left(\left\langle \boldsymbol{v}, \dot{\boldsymbol{D}}^T \right\rangle\right) \dot{\boldsymbol{Y}}$, and optimize

$$\mathcal{L}_{\mathrm{patch}} = -\sum_{l=1}^{L} m_l P^{\mathrm{patch}}(\boldsymbol{z}_l^1)^T \log\left(P^{\mathrm{patch}}(\hat{\boldsymbol{z}}_l^1)\right), \quad (4)$$

where we remove the learnable discrete tokens $\phi$ in favor of non-parametric embeddings $\boldsymbol{E}_{\mathrm{P}}$ and introduce the SEs' soft contributions through $\dot{\boldsymbol{Y}}$.

The proposed SOP-MIM task (4) encourages the network to reconstruct the missing patches so that local-level SEs from multiple SOPs produce consistent predictions between reconstructed and original embeddings.

The final loss is a convex combination of the two losses, $\mathcal{L}_{\mathrm{SOP}} = \lambda_1 \mathcal{L}_{\texttt{[CLS]}} + \lambda_2 \mathcal{L}_{\mathrm{patch}}$. By default, $\lambda_1 = \lambda_2 = 1$.

## 3. Main Experiments

We begin by assessing the quality of the pre-trained representations across a range of downstream tasks, adhering to the experimental protocol outlined by Zhou et al. (2022). Subsequently, we justify the choices in our architecture by ablating the methods' main components. For more details, please cf. Appendix A.

### 3.1. Linear Evaluation on ImageNet

$k$**-NN and Linear Probing.** In Table 1, we evaluate the linear transferability of the representations learned with SOP using two protocols: (1) non-parametric $k$-NN and (2) linear models. For the $k$-NN estimator, we tested different values of $k \in 10, 20, 100, 200$ and reported the best results. For linear probing, we used pre-trained encoders as feature extractors and trained a linear layer on top of the frozen features. On the $k$-NN benchmark, SOPs improve over existing methods across all choices of backbones. For ViT-L (307 million parameters), it improves over iBOT by $+1.2\%$ top-1 accuracy, reaching $79.2\%$, which is similar to I-JEPA's ViT-H (632 million parameters) *linear* top-1 accuracy of $79.3\%$ (Assran et al., 2023). These results suggest strong performance of SOP's *off-the-shelf* features. Additionally, we report performance values for the *supervised* baseline DeiT (Touvron et al., 2021), as well as for the strong SwinT (Liu et al., 2021) baseline EsViT (Li et al., 2022).

We observed an interesting performance scaling when training ViTs with the SOP algorithm. As we increased the complexity of the ViT backbones, the performance gains were greater than those observed for competing methods. In Table 1, while SOP 's performance using the ViT-S backbone

*Table 1.* $k$**-NN and Linear probing, semi- and full fine-tuning on ImageNet-1M.**

| METHOD | ARCH | EP. | $k$-NN | LIN. | 1% | 10% | 100% |
|---|---|---|---|---|---|---|---|
| ESVIT | SWIN-T/14 | 300 | 77.0 | 78.7 | | | |
| IBOT | SWIN-T/14 | 300 | 76.2 | 79.3 | | | |
| SOP | SWIN-T/14 | 300 | **77.2** | **79.4** | | | |
| DEIT | VIT-S/16 | 800 | 79.3 | 79.8 | | | |
| DINO | VIT-S/16 | 800 | 74.5 | 77.0 | 60.3 | 74.3 | 82.0 |
| IBOT | VIT-S/16 | 800 | 75.2 | 77.9 | 61.9 | 75.1 | 82.3 |
| MASSL | VIT-S/16 | 800 | 75.1 | 77.8 | | | |
| SOP | VIT-S/16 | 800 | **75.3** | 77.9 | **62.1** | 75.1 | 82.3 |
| DEIT | VIT-B/16 | 400 | 81.0 | 81.8 | 75.6 | 81.4 | |
| MOCO-V3 | VIT-B/16 | 400 | | 76.7 | | | |
| NNCLR | VIT-B/16 | 1000 | | 76.5 | | | |
| DINO | VIT-B/16 | 400 | 76.1 | 78.2 | 64.4 | 76.3 | 83.6 |
| IBOT | VIT-B/16 | 400 | 77.1 | 79.5 | 68.5 | 78.1 | 84.0 |
| MASSL | VIT-B/16 | 400 | 77.2 | 79.6 | | | |
| SOP | VIT-B/16 | 400 | **78.2** | **79.9** | **69.5** | **78.4** | **84.2** |
| IBOT | VIT-L/16 | 250 | 78.0 | 81.0 | | | 84.8 |
| I-JEPA | VIT-L/16 | 600 | | 77.5 | | | |
| | VIT-H/14 | 300 | | 79.3 | | | |
| SOP | VIT-L/16 | 250 | **79.2** | **81.2** | | | **84.9** |

*Table 2.* **Object detection and instance segmentation on COCO and semantic segm on ADE20k**. ViT-B encoders.

| METHOD | DET. $AP^B$ | ISEG. $AP^M$ | SEG$^\dagger$ MIOU | SEG MIOU |
|---|---|---|---|---|
| SUP. | 49.8 | 43.2 | 35.4 | 46.6 |
| BEIT | 50.1 | 43.5 | 27.4 | 45.8 |
| DINO | 50.1 | 43.4 | 34.5 | 46.8 |
| IBOT | 51.2 | 44.2 | 38.3 | 50.0 |
| SOP | **51.4** | **44.3** | **38.7** | **50.6** |

is similar to existing solutions, **more complex backbones, such as ViT-B/L and SwinT, produce larger performance gains**. These gains are primarily shown in the $k$-NN evaluation, suggesting a strong boost in the off-the-shelf representational power for retrieval tasks, cf. Section 3.5.

### 3.2. Semi-Supervised Fine-Tuning on ImageNet

In Table 1, we measure SOP's representation capacity to learn tasks using a limited set of labeled examples. Following the *unsupervised pre-train*, *supervised fine-tune* protocol, we report top-1 accuracy using $1 - 10\%$ of ImageNet-1M labeled images. We observe that SOP's performance improves and surpasses competing methods as more complex encoders are used. We observed a performance gap $(+1.0\%)$ between SOP and iBOT in smaller data regimes, such as the $1\%$ labeled data. As the fraction of annotated data increases, performances tend to level out. Following previous work (Chen et al., 2020b), we fine-tuned the pre-trained encoders for 1000 epochs from the first layer of the projection head, cf. Appendix B.

*Table 3.* **Transfer learning by fine-tuning SSL methods on smaller datasets.** Top-1 accuracy for ViT-B encoders.

| METHOD | $C_{10}$ | $C_{100}$ | INAT$_{18}$ | INAT$_{19}$ | FLWRS | CARS |
|--------|------|-------|--------|--------|-------|------|
| RAND | 99.0 | 90.8 | 73.2 | 77.7 | 98.4 | 92.1 |
| BEIT | 99.0 | 90.1 | 72.3 | 79.2 | 98.0 | 94.2 |
| DINO | 99.1 | 91.7 | 72.6 | 78.6 | 98.8 | 93.0 |
| IBOT | 99.2 | 92.2 | **74.6** | 79.6 | 98.9 | 94.3 |
| SOP | **99.3** | **92.4** | **74.6** | **79.7** | **99.0** | **94.5** |

## 3.3. Dense Prediction Tasks

To address a wide range of downstream tasks such as object detection, segmentation, and classification, an optimal fixed-size representation should balance coarse and fine-grained features. To evaluate the effectiveness of SOP in dense prediction tasks, we consider three downstream evaluations: (1) *object detection*, (2) *semantic*, and (3) *instance segmentation*. Complementary to the linear evaluations in Section 3.1, the learned representations must encode information beyond the class level, including the object's localization, shape, and ability to discriminate among different instances.

**Object Detection and Instance Segmentation on COCO.** In Table 2, the first and second columns present the AP$^b$ and AP$^m$ metrics for various SSL methods on the COCO dataset (Lin et al., 2014), with Mask R-CNN (He et al., 2017) as the task layer. The entire network is fine-tuned for 12 epochs, following the protocol outlined by Zhou et al. (2022). The representations learned using the SOP strategy show modest improvements of $+0.2$ in AP$^b$ and $+0.1$ in AP$^m$ over iBOT for object detection and instance segmentation, respectively.

**Semantic Segmentation on ADE20K.** In Table 2, the third and fourth columns report the mean intersection over union (mIoU) for semantic segmentation on the ADE20K dataset (Zhou et al., 2017). Following Zhou et al. (2022), we consider two protocols: (1) linear probing and (2) fine-tuning. For linear probing, the patch tokens from the pre-trained SOP encoder are kept fixed, and only a linear model is trained on top of the frozen features. In the fine-tuning protocol, the task layer in UPerNet (Xiao et al., 2018) is used, and we fine-tune all the parameters of the network. In both scenarios, SOP's pre-trained representations improved upon iBOT's strong baselines by $+0.4$ and $+0.6$ mIoU, respectively, and further increased the gap to the supervised baselines by $+3.3$ and $+5.0$ mIoU, respectively.

## 3.4. Transfer Learning

In Table 3, we study transfer learning tasks using SOP pre-trained encoders as initialization to perform fine-tuning on several classification tasks using smaller datasets. We report top-1 accuracy for six datasets including

*Table 4.* **Video object segmentation on DAVIS 2017.** We report mean region similarity $\mathcal{J}_m$ and mean contour-based accuracy $\mathcal{F}_m$.

| METHOD | DATA | ARCH. | $(\mathcal{J}\&\mathcal{F})_m$ | $\mathcal{J}_m$ | $\mathcal{F}_m$ |
|--------|------|-------|---------|--------|--------|
| *Sup.* | | | | | |
| IN-1K | IN-1K | VIT-S/8 | 66.0 | 63.9 | 68.1 |
| STM | I/D/Y | RN50 | 81.8 | 79.2 | 84.3 |
| *Self-Sup.* | | | | | |
| CT | VLOG | RN50 | 48.7 | 46.4 | 50.0 |
| MAST | YT-VOS | RN18 | 65.5 | 63.3 | 67.6 |
| STC | KINETICS | RN18 | 67.6 | 64.8 | 70.2 |
| DINO | IN-1K | VIT-S/16 | 61.8 | 60.2 | 63.4 |
| | IN-1K | VIT-B/16 | 62.3 | 60.7 | 63.9 |
| IBOT | IN-1K | VIT-S/16 | 61.8 | 60.4 | 63.2 |
| | IN-1K | VIT-B/16 | 62.7 | 61.7 | 63.7 |
| SOP | IN-1K | VIT-B/16 | **63.3** | **61.7** | **65.0** |

CIFAR-10/100 (Krizhevsky & Hinton, 2009), iNaturalist 2018/2019 (Van Horn et al., 2018), Oxford 102 Flower (Nilsback & Zisserman, 2008), and Stanford Cars (Krause et al., 2013). SOP pre-trained encoders demonstrate strong downstream performance on fine-tuning protocols, surpassing competitors on **5 out of 6 datasets** with modest gains. We hypothesize that the extended fine-tuning regime of 1000 epochs, as per Zhou et al.'s (2022) protocol, leads most methods to achieve similar performance levels, indicating a saturation point.

## 3.5. Image Retrieval

**Image retrieval.** To assess the image retrieval properties of SOP's representations, we consider the revisited Oxford and Paris image retrieval datasets (Radenović et al., 2018). Each dataset is divided into three sets of increasing difficulty. We use SOP's frozen encoders as feature extractors and apply $k$-NN classification on the frozen features. In Table 5, we report Mean Average Precision (mAP) for the Medium (M) and Hard (H) splits. Features pre-trained with SOP significantly outperform current state-of-the-art methods, improving mAP performance by up to $+3.2$ on the Hard split of both benchmarks. We also include results from a supervised retrieval-specific baseline (Revaud et al., 2019).

**Video instance segmentation.** In Table 4, we use frozen patch tokens from SOP's pre-trained encoders to perform video scene segmentation using a nearest neighbor classifier between consecutive frames. As we do not update any additional parameters, this evaluation is particularly valuable for assessing the fine-grained downstream capabilities of SOP's frozen features, which are learned through reconstruction using our proposed SOP-MIM task (4). We compare the performance of SOP to existing SSL methods and to a supervised ViT-S/8 model trained on ImageNet-1M. SOP's features surpass the iBOT baseline by up to $+1.3$ on mean contour accuracy $\mathcal{F}_m$.

*Table 5.* **Image retrieval.** mAP using *off-the-shelf* features.

| | | | $\mathcal{R}\mathcal{O}$x | | $\mathcal{R}$Par | |
|---|---|---|---|---|---|---|
| METHOD | ARCH. | EPO. | M | H | M | H |
| SUP. | RN101 | 100 | 49.8 | 18.5 | 74.0 | 52.1 |
| DINO | VIT-B/16 | 400 | 37.4 | 13.7 | 63.5 | 35.6 |
| IBOT | VIT-B/16 | 400 | 36.8 | 14.3 | 64.1 | 36.6 |
| MASSL | VIT-B/16 | 400 | 39.3 | 14.1 | 65.8 | 38.1 |
| SOP | VIT-B/16 | 400 | **42.7** | **17.5** | **67.3** | **41.3** |

*Table 6.* **Robustness against background changes.** ViT-B encoders.

| | BACKGROUND CHANGES | | | | | | | CLEAN |
|---|---|---|---|---|---|---|---|---|
| | OF | MS | MR | MN | NF | OBB | OBT | IN-9 |
| IBOT | 91.9 | 89.7 | 81.9 | 79.7 | 54.7 | 17.6 | 20.4 | 96.8 |
| MASSL | 91.0 | 90.2 | 83.0 | 80.4 | 53.4 | 15.8 | 23.7 | 97.6 |
| SOP | **93.3** | **91.4** | **85.6** | **83.1** | **55.8** | **19.9** | 22.8 | 97.1 |

## 3.6. Robustness

We evaluate the performance of SOP's pre-trained encoders on a robustness test containing seven variations of foreground/background mixing and masking using the ImageNet-9 dataset (Xiao et al., 2020). Results for ViT-B encoders are presented in Table 6. Our method significantly outperforms competitors in **six of the seven** background changes with notable gains in most of the categories: *Only-FG* (OF) +2.3, *Mixed-Rand* (MR) +2.6, *Mixed-Next* (MN) +2.7, and *Only-BG-B* (OBB) +2.3.

## 4. Ablations

To understand why the SOP strategy learns useful visual representations from unsupervised data, we examine its main components and the rationale for selecting the optimal set of hyperparameters. Unless otherwise specified, ablations were conducted using ViT-S encoders pre-trained for 300 epochs *without* multicrop augmentation.

**Online vs. non-parametric tokenizers.** In Table 7, we compare the performance of methods using online and pre-trained tokenizers vs. our non-parametric approach. We ablate the effect of each loss function, (2) and (4). Optimizing both loss functions, $\mathcal{L}$[CLS] + $\mathcal{L}$[MIM], our method achieves a $k$-NN top-1 accuracy gain of 0.9% over iBOT. Minimizing only the $\mathcal{L}$[CLS] objective, similar to DINO (Caron et al., 2021), SOP's $k$-NN performance improves by 1.0%. Lastly, optimizing only the $\mathcal{L}$[MIM] loss, SOP's non-parametric strategy (4) outperforms the parametric counterpart iBOT by 7.3% accuracy points in $k$-NN, indicating that the proposed SOP-MIM learns faster and contributes more to the final representation.

*Table 7.* **Parametric vs. non-parametric tokenizers.** $\Delta$: pretrained DALL-E encoder.

| Method | $\mathcal{L}$[MIM] | $\mathcal{L}$[CLS] | $k$-NN | Lin. |
|---|---|---|---|---|
| iBOT | ✓ | ✓ | 69.1 | 74.2 |
| | ✓ | ✗ | 9.5 | 29.8 |
| BEIT | $\Delta$ | ✗ | 6.9 | 23.5 |
| DINO | ✗ | ✓ | 67.9 | 72.5 |
| BEIT+DINO | $\Delta$ | ✓ | 48.0 | 62.7 |
| SOP | ✓ | ✓ | **70.0** | **74.3** |
| | ✓ | ✗ | **16.8** | 30.2 |
| | ✗ | ✓ | **68.9** | 72.8 |

*Table 8.* **Multiple SEs on pretext tasks.** $k$-NN top-1 accuracy.

| | $k$ Support Embeddings [CLS] | | | | |
|---|---|---|---|---|---|
| [MIM] | 1 | 2 | 4 | 8 | 16 |
| 1 | 69.3 | 69.6 | 69.5 | **70.0** | 69.5 |
| 2 | | 69.7 | 69.3 | 69.6 | 69.4 |
| 4 | | | 69.5 | 69.2 | 69.6 |
| 8 | | | | 69.4 | 69.3 |

*Table 9.* Modeling SEs' contributions in the SOP algorithm.

| METHOD | SOFT | ONE HOT |
|---|---|---|
| $k$-NN | **70.0** | 69.7 |

**On the number of support embeddings.** In Table 8, we examine the impact of the number of support embeddings (SEs) within an SOP on each of the proposed loss functions. Our findings indicate that the global loss (2), which operates on [CLS] tokens, benefits from the inclusion of SEs. Performance improves as the number of SEs per SOP increases, peaking at *eight*, after which performance starts to decrease. Conversely, for the local SOP-MIM loss (4), Table 8 shows that multiple SEs neither improve nor degrade performance. The empirical results in Table 10 suggest a meaningful performance gain from the incorporation of multiple SEs. We hypothesize that such gains come from the ability of SEs to better adapt to local structures in the latent space, resulting in an enhancement of the prototypes' features, which in turn benefits the SSL view optimization process.

**On the contribution of support embeddings.** As described in Section 2.2.1, SEs are selected as the closest embeddings to the SOP's anchor using spherical $k$-NN. When combining the individual scores of SEs within an SOP, the contribution of each SE to a view is proportional to its distance to the anchor. Consequently, SEs closer to the anchor have a stronger influence on the view membership calculation than those farther away. In Table 9, we explore an alternative approach to modeling SE's contributions. Instead of using the distance to the SOPs' anchors as the contribution weight, we assign a one-hot distribution to

*Table 10.* **Learning multiple SOPs.** $k$-NN top-1 accuracy.

| METHOD | 1024 | 2048 | 4096 | 8192 | 16384 |
|--------|------|------|------|------|-------|
| iBOT | 67.6 | 67.9 | 68.3 | 69.1 | 68.8 |
| SOP | **69.5** | **69.7** | **70.0** | **69.9** | **69.7** |

each SE, meaning that SEs within an SOP contribute equally when predicting the views' membership to an SOP. Our proposal demonstrates robustness to both methods, with a slight preference for our soft-contribution approach.

**On the number of Self-Organizing Prototypes.** The SOP strategy optimizes random regions in the representation space by exploring feature locality. In Table 10, we examine how the number of SOPs affects the learned representations. The results indicate that our method is robust to a varied number of SOPs. Notably, unlike prototypical SSL, which requires many prototypes, our method does not require a large number of SOPs to produce transferable off-the-shelf features. With 1024 SOPs (akin to 1024 prototypes in (Zhou et al., 2022)), we report 69.5% top-1 k-NN accuracy on ImageNet-1M, which is 1.9% above iBOT's respective performance (67.6%). Moreover, the difference between optimizing 1024 SOPs vs. the best configuration (4096) is only 0.5%. For iBOT, a similar comparison yields a larger performance discrepancy of 1.5%. These results strongly suggest that SOPs alleviates the necessity and issues associated with over-clustering in prototypical SSL.

## 5. Related Work

**Clustering and Representation Learning.** Combining clustering and deep learning has been a promising approach for unsupervised visual representation learning. Caron et al. (2018; 2019); Van Gansbeke et al. (2020) incorporated classic methods such as $k$-Means and $k$-NN in a deep unsupervised learning framework for visual features. Asano et al. (2020) proposed a self-labeling unsupervised method as an instance of the optimal transport problem. Caron et al. (2020) proposed a mini-batch version of the Sinkhorn-Knopp algorithm (Cuturi, 2013) to optimize cluster assignments between views of an image. Silva & Ramírez Rivera (2022) followed the clustering idea using SGD. Caron et al. (2021) scaled previous ideas to ViTs (Dosovitskiy et al., 2020). Inspired by modern NLP methods (Devlin et al., 2019), Zhou et al. (2022) investigated the masked image modeling (MIM) pretext task, also studied by Bao et al. (2021). These methods require special regularization techniques, such as centering, sharpening, and Sinkhorn-Knopp, to avoid ill-posed states.

**Non-parametric SSL.** The term non-parametric does not imply learning systems without parameters. Instead, it de-

scribes a framework where the relationship between variables can be derived from the data without assuming any parametric form (Sanborn et al., 2024). Wu et al. (2018) proposed a non-parametric alternative to the parametric softmax classifier to solve unsupervised classification problems at the instance level using Noise Contrastive Estimation (NCE) to approximate the full softmax. Subsequent work by He et al. (2020) and Chen et al. (2021) builds upon this idea but uses augmented versions of the same image (views) as positives. He et al. (2020) employed a memory bank to sample negative pairs and optimized a variation of the NCE loss, termed the InfoNCE (Oord et al., 2018). Chen et al. (2020a) avoided an external memory by exploring in-batch representations to sample negatives. Similarly, Dwibedi et al. (2021) optimized the InfoNCE using different images as positive pairs. For each input image, the most similar representation in memory is taken as a positive and the rest of the representations in memory are deemed as negatives. Recently, Silva et al. (2024) proposed a non-parametric approach for clustering-based SSL. The primary assumption is that views of an image should produce similar prediction patterns when compared to representations of similar images stored in memory.

**SOP.** Different from previous approaches, the proposed SOP strategy learns image embeddings by considering the viewpoints of many semantically similar support embeddings (SEs) from different images representing regions of salient features in the data. Each SOP represents an adaptable latent prototype in the data. Each SE encodes its own aspects of an SOP, and when combined, enriches the SOP's features. Our method does not require negative sampling and does not optimize the NCE or InfoNCE objectives. Moreover, our method is a general framework, and under a strict configuration, it is equivalent to the framework of Silva et al. (2024). Additionally, we propose the novel SOP-MIM (Section 2.3) loss, where the reconstruction task is based on the viewpoints of local-level SEs representing different image patches in a non-parametric space.

## 6. Conclusions

We presented Self-Organizing Prototypes, a novel SSL pretraining strategy to learn effective representations from unlabeled images. SOP addresses the problem of underrepresented prototypes in SSL by enhancing the feature set of prototypes through multiple support embeddings that reside in a semantically similar region in the non-parametric space of features. Our method avoids learning prototypes and presents two novel non-parametric pretext tasks that are stable to train and do not require extra regularization to avoid collapsed solutions. We showed that training SSL methods with the SOP strategy alleviates the problems associated with over-clustering present in the majority of current

state-of-the-art prototypical SSL. Our comprehensive benchmarking showed that SOP's visual representations perform well in many downstream tasks such as object detection, instance and semantic segmentation, image retrieval, and linear probing. Additional improvements such as hyperparameter tuning, extra regularizers, and scaling techniques, as studied by Oquab et al. (2023), can potentially improve SOP's performance and are left for future work.

## Acknowledgements

The computations were performed in part on resources provided by Sigma2—the National Infrastructure for High Performance Computing and Data Storage in Norway—through Project NN8104K. This work was funded in part by the Research Council of Norway, through its Centre for Research-based Innovation funding scheme (grant no. 309439), and Consortium Partners. This study was financed in part by the Coordenação de Aperfeiçoamento de Pessoal de Nível Superior—Brasil (CAPES)—Finance Code 001.

## Impact Statement

This paper presents work whose goal is to advance the field of Machine Learning. There are many potential societal consequences of our work, none which we feel must be specifically highlighted here.

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

# A. Implementation Details

For the main experiments in Section 3, we train SSL encoders using the SOP strategy with three Vision Transformer architectures: ViT-Small, ViT-Base, and ViT-Large, with 21, 85, and 307 million parameters, respectively. Additionally, we experiment with a Swin-Transformer backbone containing 28 million parameters. Following previous methods (Caron et al., 2021; Zhou et al., 2022), we create 12 views of the same image at each training iteration. Views indexed from $v = \{0, 1\}$ have shape $x^v \in \mathbb{R}^{224 \times 224 \times 3}$, and views indexed from $v = \{2, 3, 4, ..., 11\}$ have shape $x^v \in \mathbb{R}^{96 \times 96 \times 3}$.

The memories $E_{\mathrm{C}} \in \mathbb{R}^{N_C \times d}$ and $E_{\mathrm{p}} \in \mathbb{R}^{N_p \times d}$ store vector representations from global and local patches, respectively. We use $N_C = 65536$, $N_p = 8192$, and set the feature dimensionality to $d = 256$. At each iteration, we update each memory following a FIFO (first-in first-out) strategy. For $E_{\mathrm{C}}$, we select the [CLS] token representation from one of the global views and insert it into one end of $E_{\mathrm{C}}$. For $E_{\mathrm{p}}$, we randomly pick one of the local patch embeddings using a uniform distribution and insert it into one end of $E_{\mathrm{p}}$.

For the non-parametric support embedding (SE) selection algorithm (2), we uniformly sample $K = 4096$ anchors. Each anchor selects an additional $k = 8$ neighbors, resulting in a total of 9 SEs per SOP. Refer to Section 4 for additional context on the optimal number of support embeddings. After selecting SEs, we create the pseudo-dataset $\boldsymbol{D} \in \mathbb{R}^{K(k+1) \times d}$, where $K$ is the number of anchors, $k$ is the number of SEs, and $d = 256$ is the feature vector dimensionality. Likewise, the soft contributions $\boldsymbol{Y} \in \mathbb{R}^{K(k+1) \times K}$.

For the SOP-MIM loss (4), we sample $\dot{K} = 512$ anchors. As shown in Table 8, the SOP-MIM loss does not seem to benefit from multiple SEs. Thus, we use a single SE (the anchor itself), to represent a local SOP. Consequently, the pseudo-dataset and labels have shapes $\dot{\boldsymbol{D}} \in \mathbb{R}^{\dot{K} \times d}$ and $\dot{\boldsymbol{Y}} \in \mathbb{R}^{\dot{K} \times \dot{K}}$.

In practice, for the global loss (2), given $K = 4096$ anchors and $k = 8$ support embeddings, the pseudo-dataset $\boldsymbol{D}$ has shape $\mathbb{R}^{36864 \times 256}$. Likewise, for the SOP-MIM local loss (4), with $\dot{K} = 512$, the pseudo-dataset $\dot{\boldsymbol{D}}$ has shape $\mathbb{R}^{512 \times 256}$.

## A.1. PyTorch Style Pseudo-code

```
-----
memory.py
-----

class Memory(nn.Module):
  def __init__(self, K, num_anchors, num_SEs, dim, smoothing=0.1):
    self.K = K
    self.num_anchors = num_anchors
    self.num_SEs = num_SEs
    self.smoothing = smoothing
    self.memory = F.normalize(randn(dim, K), dim=0)
    self.labels = arange(0, num_anchors)
    self.labels = self.labels.repeat_interleave(num_SEs, dim=0).unsqueeze(1)

  def _select_anchor_idx(self):
    indices = multinomial(ones(self.K), self.num_anchors, replacement=False)
    return indices.unsqueeze(0)

  def forward(self, s_embeds, t_embeds):
    """
    s_embeds: [N x dim]
    t_emebds: [N x dim]
    """
    labels = self.labels
    smooth = self.smoothing
    memory = self.memory

    anchor_idx = self._select_anchor_idx()
    anchor_embeds = take_along_dim(memory, indices=anchor_idx, dim=1) # [dim x num_anchors]

    # top-k nearest neighbours
    similarities = anchor_embeds.t() @ memory
    scores, indices = topk(
```

```
            similarities, k=self.num_SEs, dim=-1)

    # get emebddings of anchors and support embeddings
    indices = indices.flatten().unsqueeze(0)
    embeddings = take_along_dim(memory, indices, dim=1) # [dim x num_SEs * num_anchors]

    smooth_value = smooth / (self.num_anchors - 1.0)
    exp_labels = full(
        (labels.shape[0], self.num_anchors), smooth_value)
    # anchors contributions sum to 1 across all SOPs
    exp_labels.scatter_(1, labels, full_like(exp_labels, 1.0 - smooth))

    s_logits = s_embeds @ embeddings / s_temp # [N x num_SEs * num_anchors]
    t_logits = t_embeds @ embeddings / t_temp # [N x num_SEs * num_anchors]

    s_probs = softmax(s_logits, dim=-1) @ exp_labels # [N x num_anchors]
    t_probs = softmax(t_logits, dim=-1) @ exp_labels # [N x num_anchors]
    return s_probs, t_probs

----
main.py
----
# f(.): student encoder
# g(.): teacher encoder
# K: memory size
# D: embedding dimension
# L: sequence length (number of patches)
# V: number of views
# N: batch size

class_memory = Memory(K=65536, num_anchors=4096, num_SEs=8, dim=256)
patch_memory = Memory(K=8192, num_anchors=512, num_SEs=1, dim=256)

for images in loader:
    # student and teacher branches
    s_out, t_out = f(images), g(images[:2]) # [V*N x D], [2*N x L, D]
    s_cls, s_patch = s_out
    t_cls, t_patch = t_out

    cls_mem_out = class_memory(s_cls, t_cls)
    s_probs, t_probs = cls_mem_out

    # global [CLS] loss
    s_probs = s_probs.chunk(V)
    t_probs = t_probs.chunk(2) # there are two global views
    cls_ce = cross_entropy(s_probs, t_probs)

    # SOP-MIM patch loss
    patch_mem_out = patch_memory(s_patch, t_patch)
    sp_probs, tp_probs = patch_mem_out

    sp_probs = sp_probs.chunk(2)
    tp_probs = tp_probs.chunk(2)
    patch_ce = cross_entropy(sp_probs, tp_probs, masks)
    loss = cls_ce + patch_ce

    optimizer.zero_grad()
    loss.backward()
    optimizer.step()

    # update memories
    update_mem(class_memory, t_cls)
    update_mem(patch_memory, t_patch)
```

*Table B.1.* Training time and memory: We report top-1 $k$-NN accuracy on ImageNet-1M, training time (hours), and memory (gigabytes) for SSL methods using ViT-S/16 backbones.

| | 100 EPOCHS | | 300 EPOCHS | | 800 EPOCHS | | |
| | $k$-NN | TIME | $k$-NN | TIME | $k$-NN | TIME | MEM |
|---|---|---|---|---|---|---|---|
| DINO | 69.7 | 24.2H | 72.8 | 72.6H | 74.5 | 180.0H | 15.4GB |
| IBOT | 71.5 | 24.3H | 74.6 | 73.3H | 75.2 | 193.4H | 19.5GB |
| MASSL | 72.7 | 24.2H | 74.7 | 72.4H | 75.1 | 177.3H | 15.1GB |
| SOP | 72.8 | 24.4H | 74.7 | 73.3H | 75.2 | 193.5H | 19.4GB |

# B. Extended Experiments

## B.1. Time and Computing Trade-off

In Table B.1, we explore trade-offs between parametric and non-parametric SSL. Following the protocol from (Silva et al., 2024), we report the training time and memory requirements for SOP and existing solutions. The main difference between iBOT/DINO and SOP is the absence of learnable prototypes in SOP. Instead, SOP employs two memory components, $E_C$ and $E_p$, to store [CLS] and patch-level representations, respectively. In contrast, iBOT learns two separate sets of prototypes: one for [CLS] tokens and a second for patch-level tokens trained with MIM. From a resource perspective, learning the prototypes requires extra memory to store gradients for updating the prototypes during the backward pass. SOP on the other hand, updates the prototypes following a simpler FIFO strategy. Despite this, the general computing time and memory requirements for pre-training SOP on ImageNet-1M are very similar to those of iBOT.

## B.2. Semi-Supervised Evaluations with Frozen Features

In Table 1, we assessed the semi-supervised performance of SSL methods using the *unsupervised pre-train* and *supervised fine-tune* paradigm. Additionally, in Table B.2, we compare the performance of multiple SSL methods on a semi-supervised task setup using frozen, *off-the-shelf* features on the ImageNet dataset. We report $k$-NN top-1 accuracy for the best-performing value of $k \in \{10, 20, 100, 200\}$ using the data splits provided by Chen et al. (2020a).

SOP 's performance significantly improves as model complexity increases. For ViT-S backbones, SOP performs comparably to iBOT in both data regimes. However, with the more complex ViT-B and SwinT backbones, the performance gap between SOP and its competitors widens significantly, with gains of $+2.3$ and $+1.4$ for ViT-B in data regimes of 1-10% labels, respectively.

We emphasize the still substantial gap between supervised methods (Touvron et al., 2021) and unsupervised methods on retrieval-based tasks. Specifically, for low data regimes, the existing gap suggests that current SSL methods still have room for improvement.

*Table B.2.* Semi-supervised evaluations with frozen features on ImageNet-1M: We report $k$-NN top-1 accuracy using 1-10% of labels. For reference, we include results from supervised DeiT (Touvron et al., 2021).

| METHOD | ARCH. | 1% | 10% |
|---|---|---|---|
| *Supervised* | | | |
| DEIT | VIT-S/16 | 77.3 | 78.7 |
| DEIT | VIT-B/16 | 80.2 | 80.9 |
| *Self-supervised* | | | |
| DINO | VIT-S/16 | 61.3 | 69.1 |
| | VIT-B/16 | 63.6 | 71.0 |
| IBOT | VIT-S/16 | 62.3 | 70.1 |
| | VIT-B/16 | 66.3 | 72.9 |
| | SWINT-14 | 64.2 | 71.5 |
| SOP | VIT-S/16 | 62.2 | 70.3 |
| | VIT-B/16 | **68.6** | **74.3** |
| | SWINT-14 | 65.3 | 72.3 |

## B.3. Dense Prediction Tasks

In Table B.3, we provide additional metrics for object detection, instance segmentation, and semantic segmentation evaluations using SOP's ViT-B pre-trained backbone. For object detection and instance segmentation, we use the Cascade Mask R-CNN as the task layer and the COCO dataset (Lin et al., 2014). In addition to the metrics reported in Table 2, we include $AP_{50}^b$ and $AP_{75}^b$ for object detection, and $AP_{50}^m$ and $AP_{75}^m$ for instance segmentation.

For semantic segmentation on ADE20k (Zhou et al., 2017), we follow the protocol from Zhou et al. (2022) and consider two scenarios: (1) training a linear layer on top of the frozen encoder, and (2) using UPerNet as the task layer.

*Table B.3.* Additional results for object detection, instance segmentation, and semantic segmentation using ViT-B encoders.

| METHOD | DET. & INST. SEG. W/ CASCADE MASK R-CNN | | | | | | SEG. W/ LIN. | | SEG. W/ UPERNET | |
|---|---|---|---|---|---|---|---|---|---|---|
| | $AP^B$ | $AP_{50}^B$ | $AP_{75}^B$ | $AP^M$ | $AP_{50}^M$ | $AP_{75}^M$ | MIOU | MACC | MIOU | MACC |
| SUP. | 49.8 | 69.6 | 53.8 | 43.2 | 66.6 | 46.5 | 35.4 | 44.6 | 46.6 | 57.0 |
| DINO | 50.1 | 68.5 | 54.6 | 43.5 | 66.2 | 47.1 | 27.4 | 35.5 | 45.8 | 55.9 |
| iBOT | 51.2 | 70.8 | 55.5 | 44.2 | 67.8 | 47.7 | 38.3 | 48.0 | 50.0 | 60.3 |
| SOP | **51.4** | **70.9** | 55.5 | **44.3** | **68.0** | **47.8** | **38.7** | **48.1** | **50.6** | **60.5** |

# C. Extended Ablations

## C.1. Multiple Tasks Improve the Learned Representations.

As explained in Section 2.2, the SOP strategy first samples a subset of anchors $A = \{a_i\}_{i=0}^K \subset E_C$, where each anchor can be seen as a hidden local cluster within the data. Next, each anchor $a_i$ selects additional representatives (support embeddings) using the $k$-Nearest Neighbor method. Thus, each SOP is represented by its anchor $a_i$ along with $k$ additional support embeddings $e_j$, as determined by the $k$-NN criterion.

This process can be repeated *multiple* times within each training iteration. In Table C.1, we report the effects of this strategy for each of the pretext tasks described in Section 2.2.1 and Section 2.3. We observe a positive trend as the number of SOP tasks performed per training iteration increases. Moreover, Table C.1 suggests that both global and local tasks benefit from this strategy.

*Table C.1.* The effect of the number of independent pretext tasks per iteration.

| | # OF TASKS [CLS] | | |
|---|---|---|---|
| [MIM] | 1 | 2 | 4 |
| 0 | 68.0 | 68.5 | 68.6 |
| 1 | 69.6 | 70.0 | 69.7 |
| 2 | | 69.7 | 69.8 |
| 4 | | | **70.0** |

## C.2. Learning Global-Level Features: [CLS] vs. Average Patch Embeddings.

In Table C.2, we investigate common strategies for learning class-level representations with ViT backbones. We compare (i) the default approach, which uses a dedicated [CLS] token to learn global information, with (ii) an alternative that averages the patch-level embeddings. For SSL pre-training with SOP, the default [CLS] token strategy yields slightly better $k$-NN performance.

*Table C.2.* Global-level representations as [CLS] vs AVG. patch visual embeddings.

| | [CLS] | AVG. PATCH |
|---|---|---|
| $k$-NN | **70.0** | 67.8 |

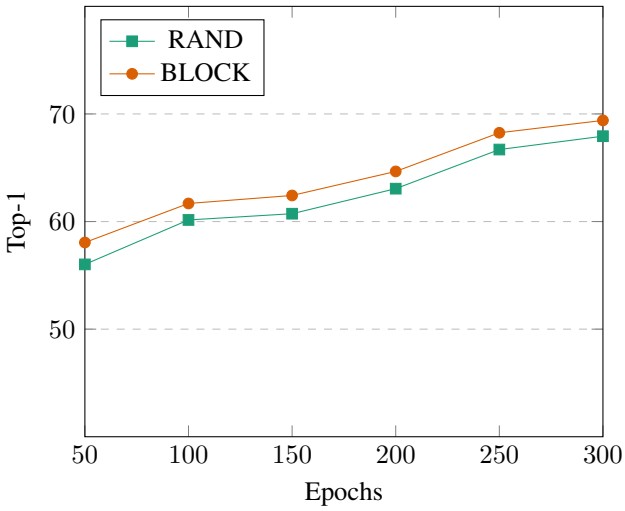

*Figure C.1.* Blockwise vs. Random masking.

*Table C.3.* Does the number of local-level anchors matter?

| $\dot{K}$ | 256 | 512 | 1024 |
|---|---|---|---|
| $k$-NN | 69.0 | **70.0** | 69.5 |

*Table C.4.* The effect of the momentum hyper-parameter on the teacher encoder.

| $m$ | .992→1 | .994→1 | .996→1 |
|---|---|---|---|
| $k$-NN | 69.2 | **70.0** | 69.6 |

### C.3. The Masking Strategy

In Figure C.1, we compare two masking strategies for the SOP-MIM task: blockwise and random masking. The blockwise approach follows the iterative technique of Bao et al. (2021), where, at each iteration, a randomly sized and shaped block of the image is masked until the desired masking ratio is reached. In contrast, the random masking strategy independently masks patches according to a specified ratio. We use a masking ratio of 0.3 (30%) for blockwise masking and 0.7 (70%) for random masking. Figure C.1 shows that the blockwise strategy consistently yields higher top-1 $k$-NN accuracy

### C.4. Does the Number of Local-Level Anchors Matter?

Similar to the global `[CLS]` task, SOP-MIM samples a subset of patch-level anchors $\dot{A}$ from the memory $E_P$, which stores local embeddings from previous iterations. Each anchor represents a local structure in the embedding space, grouping patch-level representations that share semantic features. In Table C.3, we examine how the number of sampled local anchors affects the $k$-NN performance of the learned representations. Overall, SOP demonstrates robustness across a wide range of sampling sizes.

### C.5. The Momentum Encoder

As is standard practice in SSL, SOPs are trained with two sibling encoders in a teacher-student setup. The student encoder is updated via gradient descent, while the teacher encoder is updated using a moving average of the student's weights: $\Phi_t = m\Phi_t + (1-m)\Phi_s$, where $\Phi_s$ and $\Phi_t$ denote the weights of the student and teacher encoders, respectively. This framework can also be interpreted from a distillation perspective, where the teacher distills knowledge from previous iterations into the student. Here, $m$ controls the flow of distillation between the two encoders and follows a cosine schedule. In Table C.4, we study the effect of the hyperparameter $m$ on the downstream performance of the learned representations.

### C.6. `[CLS]` and Patch Memory Sizes

**`[CLS]` Memory Size:** In Table C.5, we ablate the effect of memory size $N_C$ on the learned representations. We observe an inverse U-shaped relationship between memory size $N_C$ and downstream $k$-NN performance: increasing $N_C$ improves

*Table C.5.* Increasing the memory size $N_C$ benefits the learned representations.

| $N_C$ | 8192 | 16384 | 32768 | 65536 | 98304 |
|---|---|---|---|---|---|
| | 67.0 | 67.7 | 68.5 | **70.0** | 69.7 |

*Table C.6.* Increasing the memory size $N_p$ benefits the learned representations.

| $N_p$ | 1024 | 2048 | 4096 | 8192 | 16384 |
|---|---|---|---|---|---|
| | 68.3 | 68.7 | 69.2 | **70.0** | 69.1 |

*Table C.7.* Anchor selection strategies for Self-Organizing Prototypes (SOPs). We compare selecting anchors **randomly** at each iteration (default SOP) with keeping anchors **fixed** during pre-training. Fixed anchor selection leads to pre-training collapse.

| FIXED ANCHORS | RANDOM ANCHORS |
|---|---|
| - | **70.0** |

performance up to a point, after which further increases lead to diminishing returns. For this experiment, the patch memory size is fixed at $N_p = 8192$.

**Patch Memory Size:** In Table C.6, we ablate the effect of patch memory size $N_p$ on the learned representations. Similar to Table C.5, we observe an inverse U-shaped curve between $N_p$ and downstream $k$-NN performance, suggesting an optimal memory size at $N_p = 8192$. The [CLS] memory size is fixed at $N_C = 65536$.

### C.7. Collapse Analysis

To assess the importance of randomization as a regularizer to prevent training collapse, we explore two algorithmic variations for selecting anchors for SOPs. As described in Section 2.2, the *default* SOP strategy selects anchors uniformly at random. In Table C.7, we compare our default strategy with an alternative anchor selection method in which anchors are randomly selected at the beginning of training and kept fixed throughout. Keeping anchors fixed during training corrupts the learned features and leads to a collapse, suggesting that randomization in anchor selection is a key ingredient for avoiding collapse.

### C.8. t-SNE Feature Visualization.

To qualitatively evaluate the features learned using our proposed non-parametric SOP strategy, Fig. C.2 and Fig. C.3 present t-SNE visualizations of features on the CIFAR-10/100 datasets. For both datasets, we compare the feature spaces learned by our method (left) and by iBOT (right), using pre-trained ViT-B encoders.

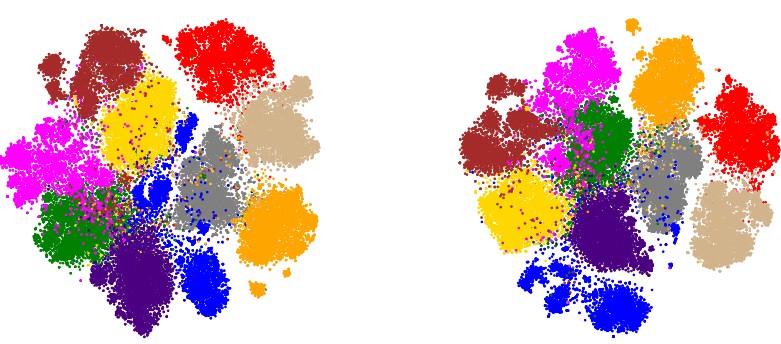

*Figure C.2.* t-SNE visualizations on CIFAR-10 with SSL pre-trained ViT-Base feature extractors: SOP (left) vs. iBOT (right).

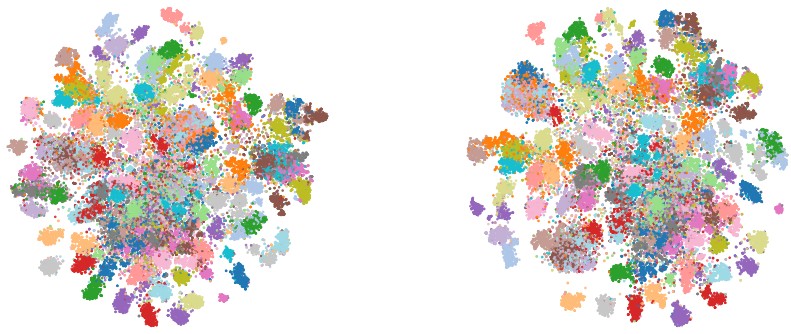

*Figure C.3.* t-SNE visualization on CIFAR100 using SSL pre-trained ViT-Base encoders as feature extractors. Qualitative results for SOP (left) and iBOT (right).

