# OpenReview forum: "Self-Organizing Visual Prototypes for Non-Parametric Representation Learning"
_ICML.cc/2025/Conference — ICML 2025 poster_

### Official Review · Reviewer_P59u · 2025-03-08

**Overall Recommendation:** 3

**Summary:**

This paper introduces ​Self-Organizing Prototypes (SOP), a non-parametric self-supervised learning (SSL) framework designed to address limitations in traditional prototypical SSL methods. Conventional approaches rely on static, learnable prototypes to represent clusters in unlabeled data, often leading to over-clustering and incomplete feature space coverage. SOP innovates by dynamically aggregating ​multiple support embeddings (SEs)—semantically similar representations stored in memory banks—to form enriched, adaptive prototypes. These SEs collectively capture complementary features within local regions of the feature space, enabling broader and more robust representations.

**Claims And Evidence:**

The claims in the submission are ​partially supported by empirical evidence but lack rigorous validation for several critical assertions.
1. "SOP alleviates over-clustering in prototypical SSL". The paper does not demonstrate that SOP achieves comparable/better performance with ​fewer prototypes than baselines (e.g., iBOT). Over-clustering is defined as using K≫C, but SOP’s default K=4,096 anchors are not justified as "non-overclustered." Besides, there is no t-SNE/UMAP visualization or entropy metric to quantify whether SOP prototypes cover the feature space more comprehensively than parametric prototypes.
2. "SOP is stable and requires no explicit regularization (e.g., Sinkhorn-Knopp)." The FIFO-based memory might propagate noise during early training stages, but no experiments address this. Besides, there is a lack of explanation on how the model can avoid training collapse.
3. "SOP-MIM improves local feature learning for dense tasks." The paper does not ablate the SOP-MIM component to isolate its contribution. For example, training with only $L_{[CLS]}$ versus full $L_{SOP}$.
4. "Non-parametric SSL is feasible without learnable prototypes." While SOP avoids explicit prototype parameters, the memory bank $E_C$/$E_P$, effectively acts as a dynamic, non-differentiable parameter store. This is a semantic distinction rather than a fundamental shift. Besides, this paper does not explain how SOP’s non-parametric design differs from parametric methods in terms of optimization dynamics or generalization.

**Essential References Not Discussed:**

No

**Experimental Designs Or Analyses:**

1. The impact of memory size ($N_C$ =65,536, $N_p$=8,192) is not tested. Larger/smaller memory banks could reveal performance trade-offs.
2. The paper uses K=4,096 anchors for SOP—similar to iBOT’s prototype count (K=1,024–10,000). No experiments show SOP achieving comparable performance with fewer prototypes.
3. While SOP claims to avoid collapse without regularization (e.g., Sinkhorn-Knopp), there is no analysis of prototype collapse (e.g., percentage of "dead" prototypes) or comparison to parametric methods.

**Methods And Evaluation Criteria:**

Yes

**Other Comments Or Suggestions:**

No

**Other Strengths And Weaknesses:**

No

**Questions For Authors:**

No

**Relation To Broader Scientific Literature:**

The paper bridges non-parametric contrastive learning (e.g., MoCo, NNCLR) and prototypical SSL (e.g., SwAV, iBOT) by introducing dynamic self-organizing prototypes (SOPs) constructed from memory-based support embeddings, extending masked image modeling (MIM) to a non-parametric regime while addressing over-clustering and scalability limitations of prior prototype-based methods.

**Theoretical Claims:**

NA

---

> ### Author Rebuttal · Authors · 2025-03-30
>
> We greatly appreciate the reviewer’s careful assessment and constructive feedback on our work. Your insights have helped us improving the manuscript. Below, we address each point in turn and provide additional visualizations and tables through anonymous links as requested.
>
> > [...] paper does not demonstrate that SOP achieves comparable performance with fewer prototypes [...]
>
> Due to space limitations, we kindly refer the reviewer to Table 10, and to our first response to reviewer KMMU, where we address a similar question.
>
> >  Besides, there is no t-SNE/UMAP visualization [...].
>
> As suggested, we provide additional t-SNE visualizations of features using pre-trained SOP and iBOT encoders.
> - [tSNE Cifar10](https://shorturl.at/W9T2h)
> - [tSNE Cifar100](https://shorturl.at/R8H9q)
>
> > The FIFO memory might propagate noise [...] lack of explanation on how the model avoids collapse.
>
> We agree with the reviewer on the importance of discussing collapse avoidance. We hypothesize that one key aspect contributing to the stability of our method is the built-in randomization of our SOPs. As quoted in line 67 and 82: “Our proposal organizes the feature space by exploring random local structures within the data”.
>
> We agree that the FIFO updates might introduce biases where newer representations are located at one end of the memory while older ones are located at the other end. **However, the SOP's built-in randomization, which selects random anchors at each iteration, breaks this “recency correlation” in the FIFO memory, preventing the bias from affecting the loss function.**
>
> To support our hypothesis, we conducted a [Collapse Analysis](https://shorturl.at/StDnj) comparing the effectiveness of our method when SOPs are randomly created at each iteration (default behavior) versus a version where SOPs are fixed after the initial random selection, similar to learnable prototypes. As shown, fixing the SOPs lead to collapse.
>
> >  [...] does not ablate the SOP-MIM [...] to isolate the contribution of the losses.
>
> We kindly refer the reviewer to Section 4: Online vs. Non-Parametric Tokenizers and Table 7, where we ablate and isolate the contribution of each proposed loss function, as suggested. In summary, training with $L_{CLS}$ alone produces good representations and surpasses competirors in $k$-NN and linear probing. Similarly, training with the non-parametric $L_{MIM}$ alone improves over iBOT (MIM alone) by $+7.3\%$ ($k$-NN).
>
> > [...] memory bank $E_C/E_p$ acts as a [...] non-differentiable parameter store. [...] a semantic distinction [..] does not explain how SOP’s design differs [...]
>
> We respectfully offer a different perspective from the reviewer on this point. We believe that learning prototypes from scratch and optimizing a memory of non-parametric representations that self-organize, have important differences, which we highlight below:
>
> 1- The optimization task does not involve gradients to update prototypes.
>
> 2- In prototypical SSL, centroids are initialized at random locations and move in the space to find an optimal position. The SOP strategy optimizes random regions in the feature space using semantically related embeddings to represent their regions.
>
> 3- Our method does not require additional terms in the loss function to avoid collapse, which is a critical component in current prototypical SSL.
>
> Regarding differences in optimization dynamics, we discuss how the proposed approach differs from prototypical SSL in terms of learning fixed prototypes vs. dynamic SOPs, lines 035, 082. About generalization, we discuss the issue of over-clustering that current methods employ and how it can hurt the learned representations (lines 052, 064, and Table 10).
>
> > [...] memory size [...] is not tested. [...]
>
> We thank the reviewer for pointing out this flaw. Please refer to [Memory Size ablations](https://shorturl.at/xYg9a) where we ablated the sizes of the $\texttt{[CLS]}$ memory $N_c$ and the size of the patch memory $N_p$. We observed a similar reversed “U” shape in both components. In general, increasing the memory size is beneficial up to a certain point, $N_c=65536$ and $N_p=8192$ for the class and patch memories, respectively.
>
> > [...] there is no analysis of prototype collapse [...].
>
> We thank the reviewer for the thoughtful suggestion. However, in our setup, it doesn’t make sense to analyze the prototypes since our working memory is different from the traditional prototypes in existing methods. The memory that we use to compute the SOPs is ever changing due to its FIFO property. Thus, SOPs are dynamic (not fixed) data structures. They are randomly created at each iteration from random locations in the feature space. Hence, we don’t have access to them as we would have access to prototypes in regular prototypical SSL. Please refer to [Collapse Analysis](https://shorturl.at/StDnj) where we show that randomly creating SOPs is the key to avoid collapse.

---

### Official Review · Reviewer_KMMU · 2025-03-09

**Overall Recommendation:** 2

**Summary:**

1-1. The authors proposed a non-parametric design called Self-organizing visual embeddings (SOLVE) in the field of self-supervised learning. Specifically, SOVE assumes the presence of multiple anchors that together represent a single concept, combining them to form a single soft label for self-supervised learning. Additionally, it presents a non-parametric design for masked image modeling loss, intended for joint use during training.

**Claims And Evidence:**

2-1. In the introduction, Lines 56-63, the issues of underrepresentation and overclustering are mentioned, but there is no supporting evidence to demonstrate whether the proposed method addresses these issues.

2-2. In Table 9, there appears to be no significant difference between the one-hot approach and the proposed soft approach. In Line 210, the effectiveness of the soft approach is explained as 'potential noise due to false positives from the SE selection algorithm,' but there is no supporting evidence provided for this claim.

2-3. According to Table 12 of [1] mentioned in the claim on Line 41, there seems to be no significant performance difference even when the number of prototypes $C=K=1000$, compared to the $1000=C <K=3000$ used by the authors of [1]. However, there is no explanation provided for this.

[1] Unsupervised Learning of Visual Features by Contrasting Cluster Assignments (NeurIPS’20)

**Essential References Not Discussed:**

8-1. It seems that there are no essential references that have not been discussed.

**Experimental Designs Or Analyses:**

5-1. In Line 269, it is mentioned that "more complex backbones produced larger performance gains," but the intuition behind this statement is not provided.

**Methods And Evaluation Criteria:**

3-1. The explanation of the proposed method is difficult to read. For example,

3-1-1. In Line 138, $N_c$ and $N_p$ are not defined.

3-1-2. In Line 197, there is a sentence starting with "we introduce $Y$," but it is only possible to infer what $Y$ is, its size, and its meaning from the following context.

**Other Comments Or Suggestions:**

None.

**Other Strengths And Weaknesses:**

10-S1. The non-parametric approach to masked image modeling appears to be a reasonable method.

10-W1. Looking at the tables provided in the main experiments, aside from image retrieval, the overall performance improvement compared to other self-supervised methods (e.g., iBOT) seems marginal.

10-W2. Upon reviewing Table B.1, the advantages in terms of train time, memory, and performance over the existing method, MaSSL, are not clearly evident.

**Questions For Authors:**

12-1. When comparing the case where "allows a SE to belong to more than one SOP" (mentioned in Line 190) is applied versus when it is not, is there a performance gain? Or was this method used simply because it is a simpler way to assign soft probabilities?

12-2. Please also refer to questions 2, 3, 5, and 10.

---

**After Rebuttal**

While the method proposed by the authors appears to be a valid approach, I remain unconvinced by the claimed advantages based on the current experimental results and the rebuttal. Therefore, I would prefer to keep my score unchanged at this point.

**Relation To Broader Scientific Literature:**

7-1. Self-supervised representation learning is an actively researched field. While various methods have been proposed, representation collapse remains a critical issue that needs to be addressed. This work proposes a new solution to this problem.

**Theoretical Claims:**

4-1. The theoretical proof is not provided in the paper.

---

> ### Author Rebuttal · Authors · 2025-03-30
>
> We sincerely thank the reviewer for his/her careful reading and constructive feedback. The insights have significantly contributed to improving our manuscript. Below, we address each point in turn.
>
> >2-1 underrepresentation and overclustering [...] no evidence [...]
>
> We argue that over-clustering and underrepresentation of prototypes are related. As noted in lines 036 and 052 (col 2), due to the weak supervision in SSL, prototypical methods tend to learn a large number of prototypes, dividing the feature space into many smaller sections, each with a prototype as its centroid. Consequently, each section contains fewer mages, biasing the prototypes towards learning simpler features, potentially leading to an underrepresentation of its region.
>
> To support our claim, we conducted an ablation study on different values of K (number of SOPs). If our claim is correct, our method should perform well with a small number of SOPs. In Table 10 and line 394, using K=1024 anchors (similar to ImageNet classes), the SOP strategy outperforms iBOT, which learns K=8192 prototypes. Furthermore, the performance difference between K=1024 and our default value of K=4096 is minimal (0.5%). In contrast, iBOT exhibits a 3x larger discrepancy (1.5%) for the same comparison; DINO [2] reports an even greater difference (1.9%).
>
> >2-2.[...] no difference one-hot vs. soft approach [...]
>
> In Table 9, we compared two strategies for modeling the contribution of SEs towards their SOPs. We reached the same conclusion as the reviewer and demonstrated that our method is robust to both strategies (line 377, column 2).
> We believe the "soft assignment" approach has a slight advantage because it handles uncertainty better, which is crucial in SSL. Intuitively, we can relate to the "label smoothing" regularizer commonly used in semi-supervised setups, which improves generalization by softening ground-truth labels, reducing noise and preventing overfitting.
>
> >2-3. no performance difference for C=K=100 in [1] [...]
>
> If we understand correctly, the reviewer is asking why SwAV [1] does not report a significant performance difference between learning K=1000 and K=3000 prototypes, which might contradict our claim. We believe there are two reasons. First, the difference between K=1000 and K=3000 may not be large enough to highlight the discrepancy. Second, performance-wise, iBOT and DINO are more effective than SwAV. In fact, DINO is the successor of SwAV (same main author), and **one of the reasons why it performs better is over-clustering**. This is demonstrated in the DINO paper [2], page 16, where there is a significant difference in performance (1.9%) from learning K=1024 to K=65536 prototypes, supporting our claim. The same pattern occurs in iBOT, (cf. Table 10). We agree that the reference to [1] should be removed from this context.
>
> [2] Emerging Properties in Self-Supervised Vision Transformers (ICCV)
>
> >3-1-1. Nc and Np not defined.
>
> Thanks for pointing this inconsistency. Specifically, $N_C$ and $N_p$ are the sizes of the memories $E_C$ and $E_p$.
>
> >3-1-2. [...] not possible to infer Y.
>
> Thank you for the feedback. The manuscript now includes the shape of $Y$ in the text. Please note that the shape of $Y \in \mathbb{R}^{K(k+1) \times K}$ is defined in Figure 2's caption, and its practical size is detailed in Sec. A, appendix. Please refer to line 193 for an explanation of $Y$, which encodes the contributions of SEs, towards their SOPs.
>
> >5-1."complex backbones, larger performance gains" no intuition [...]
>
> Thank you. The manuscript has been updated with an intuitive explanation. We believe the performance gain from larger backbones is linked to a higher efficiency of the SOP mechanism, which provides richer features to optimize views. Since SOPs use support embeddings (SEs) to represent their region in latent space, complex backbones can better utilize SEs' features, resulting in larger gains (cf. Table 1, line 264).
>
> >10-W1. overall improvements [...] seem marginal.
>
> Due to space limitations, we kindly refer the reviewer to our third response to reviewer Rjpz, where we discuss this issue in detail.
>
> >10-W2. Table B.1, advantages [...] not clearly evident
>
> While we did not claim advantages in computing and memory, Table B.1 shows that our non-parametric approach does not incur additional computational/time requirements and can be trained on commodity hardware. Please, note that MaSSL does not use a MIM loss, reducing computing/time resources.
>
> >12-1. When comparing the case where [...]
>
> The fact that a support embedding (SE) may belong to more than one SOP is a consequence of the k-NN algorithm. Each anchor selects the k closest embeddings as SEs, allowing distinct anchors to select the same SE. To address the reviewer's concerns, we tested a [Non-shared SEs] (https://shorturl.at/iqdmT) version, where an SE cannot participate in more than one SOP. Results show no practical difference, likely due to the small number of such cases.

---

### Official Review · Reviewer_Rjpz · 2025-03-11

**Overall Recommendation:** 2

**Summary:**

This paper proposes a new self-supervised learning approach that leverages multiple anchors and support embeddings to infer soft label for self-supervised learning. Extensive experiments demonstrate that the proposed method achieves favorable performance across various downstream tasks.

** Update after rebuttal **

The performance advantages and novelty of this paper are not evident compared to existing SSL works. Thus I decide to maintain my rating.

**Claims And Evidence:**

The paper’s writing is poor, and not easy to follow.

**Essential References Not Discussed:**

No

**Experimental Designs Or Analyses:**

Yes

**Methods And Evaluation Criteria:**

Yes

**Other Comments Or Suggestions:**

Some typo issues. The format of double quotes is wrong, e.g., line 076 “region”.

**Other Strengths And Weaknesses:**

Strengths

- The paper conducts thorough experiments on various downstream tasks.
- This paper proposes self-organizing visual prototypes as a non-parametric prototypical SSL technique.



Weaknesses

- Figure 2 lacks some illustration details. It is recommended to add explanations below the figure clarifying the meanings of squares, circles, and different colors to improve readability and aid reader comprehension.

- What are the advantages of SOP over learnable tokens? There is no discussion regarding the trade-offs between non-parametric and parametric methods.

- The authors conduct experiments on various downstream tasks. However, according to the experimental results, the proposed method shows only marginal improvements over previous baselines, with some results improving by less than one percentage point, which may fail to convincingly demonstrate the superiority of the proposed method.

- In previous self-supervised learning methods, a view typically means an augmentation of the input image. However, in the context of SOP,  the specific meaning of “view” is not clear. While the authors mention that “each support embedding predicts the degree of similarity (or likelihood) of a view to its ”region” in latent space (SOP)”, this description needs more clarity.

**Questions For Authors:**

As SOP is a non-parametric prototypical learning, does it have higher computational efficiency compared to previous methods? The inclusion of an efficiency comparison is recommended.

**Relation To Broader Scientific Literature:**

See Other Strengths And Weaknesses.

**Theoretical Claims:**

Yes

---

> ### Author Rebuttal · Authors · 2025-03-29
>
> We thank the reviewer for his/her careful assessment and constructive feedback, which helped us clarify key concepts about our work, improve the readability of the paper, and strengthen our arguments. Below, we address each of the reviewer's points in turn.
>
> > Figure 2 lacks some illustration details [...] add explanations [...] for squares, circles, and different colors [...].
>
> We thank the reviewer for the constructive feedback and apologize for the lack of clarity in Figure 2. To improve readability and aid reader comprehension, we have added text labels for each symbol in Figure 2. Please refer to [Figure 2](https://shorturl.at/hlvGW). The updated figure now includes labels below each core component, clarifying the meanings of squares, circles, and different colors.
>
> > What are the advantages of SOP over learnable tokens? no discussion of trade-offs [...]
>
> In the paper, we highlight the following advantages of our method over existing prototypical SSL approaches:
>
> 1- Our work reaffirms the feasibility of non-parametric SSL. Our goal is not to discourage parametric clustering-based approaches. Instead, we sought to demonstrate that our proposed non-parametric method is a viable alternative, a different path that can contribute to the scientific discussion and development of visual unsupervised learning.
>
> 2- Our method avoids learning prototypes from random weights and exhibits several key benefits:
> - Stability: The method is inherently stable.
> - No Extra Regularizers: It does not require additional regularizers to prevent mode collapse (Section 2.2.1).
> - Extensibility: It is extensible to various pretext tasks such as MIM (Section 2.3 and Table 7).
> - No Over-Clustering: It does not necessitate over-clustering (Section 4, Table 10).
> - Transferable Representations: The SOP pre-training strategy produces representations that are transferable across many different tasks, with varying degrees of difficulty (Tables 1 to 7).
>
> 3- Moreover, we show that SOP’s performance increases as we scale the model architecture (Section 3.1 and Table 1).
>
> > [...] the proposed method shows only marginal improvements over previous baselines, with some results improving by less than one percentage point [...].
>
> We acknowledge the reviewer's concern regarding the magnitude of improvements. However, **we believe that scoring well across a wide range of downstream tasks demonstrates the robustness, stability, and effectiveness of the proposed approach, which is particularly challenging in SSL.**
>
> As recognized, we conducted thorough evaluations on various downstream tasks, including classification, retrieval, robustness, fine-tuning, transfer learning, object detection, and segmentation. While some improvements may seem marginal, we would like to highlight several non-trivial improvements:
>
> - Image Retrieval (Table 5): mAP improved by +3.2.
> - Robustness Against Background Changes (Table 6):
>    - OnlyFG (OF) +2.3
>    - Mixed-Rand (MR) +2.6
>    - Mixed-Next (MN) +2.7
>    - Only-BG-B (OBB) +2.3
> - kNN with ViT-Large backbone (Table 1): +1.2%
> - Semi-Supervised Fine-Tuning with 1% of Labeled Data (Table 1): +1.0%
>
> Additionally, in Table B.2 (appendix), we report an improvement gain of +2.3 in semi-supervised learning (low data regime) on ImageNet using frozen features.
>
> **These results collectively demonstrate that our method provides non-trivial improvements and is effective across various tasks and settings.**
>
> > In previous SSL methods, a view means an augmentation of the input image [...] in the context of SOP, the meaning of “view” is not clear. [...], this description needs more clarity.
>
> We apologize for any confusion regarding the term "view." In this work, views are augmented versions of an image, similar to existing SSL methods. Based on the reviewer's feedback, we have improved the readability of [Figure 2]
>  (https://shorturl.at/hlvGW).
>
> Additionally, we define a view $x^v$ as an "augmented version of an image" in the Notation paragraph (page 094, column 2), and we use multiple views (multicrop) during optimization. The misunderstanding may have arisen from a part of the text where we intuitively suggest viewing SOPs as augmented prototypes. This interpretation emphasizes that, rather than relying on a single prototype, SOPs group features among different SEs, collectively representing the region in space more accurately.
>
> > Some typo issues [...].
>
> Thanks, we have corrected the typo issues and the format of double quotes.
>
> > [...] does SOP has higher computational efficiency compared to previous methods? The inclusion of an efficiency comparison is recommended.
>
> We agree with the reviewer on the importance of an efficiency comparison. Please refer to Table B.1 (appendix), where we compare the computing and time resources of our approach against existing prototypical SSL methods. Our findings indicate that the computational and memory requirements of SOP are similar to those of existing approaches such as iBOT.

---

> > ### Comment · Reviewer_Rjpz · 2025-04-07
> >
> > After reading author's response, I decide to maintain my rating.

---

> > > ### Author Response · Authors · 2025-04-08
> > >
> > > Thank you for the response.
> > >
> > > We would like to summarize this review by stating that we have given our best to address every single point in turn, either by pointing to missed experiments in the paper or providing clarifications, such as the improved visual description of Figure 2, as requested.
> > >
> > > In short:
> > > - We clarified the question regarding the benefits of our proposed approach.
> > > - We addressed the concern about "marginal" improvements by demonstrating that our method not only performs well on a variety of benchmarks but also achieves non-trivial downstream performance in important applications.
> > > - We clarified the misconception regarding views.
> > > - We fixed typos, as requested.
> > > - We directed the reviewer to the requested performance analysis present in Table B.1 (appendix).
> > >
> > > Respectfully, we believe the raised concerns are minor and should not be a reason to reject our work.
> > >
> > > We sincerely thank the reviewer for their efforts.

---

### Official Review · Reviewer_2CgY · 2025-03-13

**Overall Recommendation:** 3

**Summary:**

This paper introduces  a novel technique for self-supervised learning,  Self-Organizing Visual Prototypes (SOP), to address limitations in existing prototypical self-supervised learning (SSL) methods.
SOP enhances the feature set of prototypes by using multiple semantically similar support embeddings (SEs) to collectively represent regions in the feature space, rather than relying on a single prototype per cluster. This approach is implemented through a non-parametric framework that avoids over-clustering and mode collapse, resulting in stable and transferable representations.  The method is evaluated on the full range of standard self-supervised benchmarks, comparing against most expected methodologies and achieving good performance improvements across the board.

**Claims And Evidence:**

The paper's claims regarding the effectiveness of Self-Organizing Visual Prototypes (SOP) are well-supported by evidence. SOP's ability to mitigate over-clustering is demonstrated through strong performance with fewer prototypes compared to existing methods.

**Essential References Not Discussed:**

None

**Experimental Designs Or Analyses:**

This paper has conducted extensive experimental verification on ImageNet, including linear evaluation, semi-supervised fine-tuning, fine-tuning, and dense prediction on COCO and ADE20K, as well as image retrieval and video segmentation tasks, proving the effectiveness of the method. And ablation experiments were performed on the shutdown components mentioned in the method.

However, I think there are still some points that can be improved:

1. The training time and memory usage of SOP and baseline methods are not compared. Non-parametric memory management may increase the computational burden and affect practicality. In particular, the efficiency of k-NN search under large memory is not discussed, and there may be scalability issues.

2. Transfer learning on smaller datasets is not very meaningful and can be placed in supplementary materials.

**Methods And Evaluation Criteria:**

The proposed SOP method and evaluation criteria are well-suited for the problem of unsupervised visual representation learning. SOP addresses limitations of existing prototype-based methods through its non-parametric approach and multiple support embeddings.

**Other Comments Or Suggestions:**

I think the clarity of the writing can be improved.
The term “prototype” is used ambiguously—sometimes referring to traditional parametric prototypes (e.g., iBOT) and other times to SOP anchors.

**Other Strengths And Weaknesses:**

Strengths:
S1 This paper introduces Self-Organizing Prototypes (SOPs), replacing fixed prototypes with dynamically constructed support embeddings (SEs) in non-parametric prototype design. This challenges the traditional assumption in prototypical SSL and enhances feature diversity beyond methods like iBOT and SwAV.

S2 The paper proposes SOP-MIM, an extension of masked image modeling (MIM) that leverages non-parametric patch-level SEs instead of learned discrete tokens (e.g., BEiT). This reduces parametric bottlenecks and offers a novel perspective on representation learning.

Weaknesses:
W1 Figure 2 is not very easy to understand, and it is difficult to make a good connection between the pseudo code and Figure 2

W2 The motivation of SOP-MIM is clear and easy to understand, but it is just an incremental work of MIM. Maybe I misunderstood it. The author can explain it further.

W3 Is the key to the improvement of this paper due to the increase in the number of anchor points? As mentioned in the introduction, the current method performs best when the number of prototypes K≫C is much larger than the number of actual categories C in the data, but the method proposed in this paper also requires larger prototypes.

**Questions For Authors:**

see weaknesses

**Relation To Broader Scientific Literature:**

The key contributions of the paper SOP are closely related to several important trends and findings in the broader scientific literature on self-supervised learning (SSL), representation learning, and computer vision.

**Theoretical Claims:**

The paper does not present formal proofs or extensive theoretical claims that require rigorous mathematical verification.

---

> ### Author Rebuttal · Authors · 2025-03-29
>
> We appreciate the reviewer's fair assessment of our work. We would like to highlight key points from the review, including our "extensive experimental verification" and the accurate summary that captures the core innovation of our method. The review emphasizes our method's ability to mitigate over-clustering and its consistent performance across a wide range of benchmarks. We acknowledge the constructive feedback and address each point in turn.
>
> > Training time and memory usage of SOP and baseline methods are not compared. [...] In particular, the efficiency of k-NN search [...].
>
> Regarding the training time and memory usage of SOP against baselines, we kindly refer the reader to Section B.1 and Table B.1 in the appendix. In Section B.1, we discuss the trade-offs and differences between parametric and non-parametric approaches. To address the reviewer’s point, we found that the computing and memory requirements of SOP are similar to existing approaches such as iBOT (Table B.1). As pointed out, our method requires running k-NN to bootstrap the support embeddings. **On the other hand, our method does not have the large learnable prototypical layer present in clustering-based methods such as DINO and iBOT, which requires extra computing and memory to compute/store gradients in the backward pass**. Our method stores 256-d representations in two separate memories E_C and E_p of sizes 65536 and 8192, which are updated following an efficient FIFO strategy.
>
> > Transfer learning on smaller datasets is not very meaningful and can be placed in supplementary materials.
>
> We acknowledge the reviewer's point and share their conclusion regarding the saturation of these benchmarks, as indicated in line 326, column 1. Following the reviewer's recommendation, we will relocate Table 3 (Transfer learning on smaller datasets) to the supplementary material and insert Table B.1 (Training time and memory trade-off) into the main text.
>
> > W1 Figure 2 is not very easy to understand, and it is difficult to make a good connection between the pseudo code and Figure 2
>
> Following the reviewer's suggestion, we have updated Figure 2 with supplementary labels to improve clarity. The revised figure can be found in Figure 2 (https://shorturl.at/hlvGW).
>
> > W2 The motivation of SOP-MIM is clear [...] it is just an incremental work of MIM. [...].
>
> The majority of work on SSL that optimizes an MIM task focuses on learnable tokens, such as iBOT, BEiT, and DINOv2. Instead, **our work presents an alternative approach**. We show that a non-parametric strategy also works well for MIM, opening new possibilities for further development.
>
> > W3 Is the key to the improvement of this paper due to the increase in the number of anchor points? [...] current method performs best when the number of prototypes K≫C is much larger than the number of actual categories C [...], but the method proposed in this paper also requires larger prototypes.
>
> Thanks for the feedback. We argue that the SOP strategy is a key factor contributing to the performance of our method. As shown in Table 8, overall performance improves with an increase in the number of support embeddings (SEs). This supports our claim that the SOP strategy, which involves bootstrapping semantically similar embeddings to better represent a random region in the feature space, is more effective than learning a single prototype for that region, as prototypical methods do.
>
> Regarding the number of anchors, we used a default value of K=4096. Although this may seem large, Table 10 demonstrates that even with K=1024 (nearly the same number of classes as in the ImageNet dataset), **our method still surpasses iBOT, which learns K=8192 prototypes**. Additionally, the performance difference between using K=1024 and K=4096 is minimal (only 0.5%). In contrast, iBOT shows a significantly larger gap (1.5%) when comparing similar settings, as detailed in line 397. These results corroborate our claims about overclustering and how the SOP strategy mitigates it.

---

### Decision · Program_Chairs · 2025-05-01

**Decision:**

Accept (poster)

**Comment:**

## Summary

The paper introduces Self-Organizing Visual Prototypes (SOP), a non-parametric self-supervised learning (SSL) technique that improves the feature set of prototypes by using multiple semantically similar support embeddings (SEs) to represent regions in the feature space. This approach avoids over-clustering and mode collapse, resulting in stable and transferable representations. The method is evaluated on standard self-supervised benchmarks and shows favorable performance across various tasks. The authors also propose a non-parametric design for masked image modeling loss, intended for joint use during training.

## Strengths
- Introduces Self-Organizing Prototypes (SOPs) replacing fixed prototypes with dynamically constructed support embeddings.
- Challenges traditional assumptions in prototypical SSL, enhancing feature diversity.
- Proposes SOP-MIM, an extension of MIM using non-parametric patch-level SEs.
- Conducts thorough experiments on various downstream tasks.

## Weaknesses
 -Figure 2 is challenging to connect with pseudo code and lacks illustrations for better readability.
- SOP-MIM is an incremental work of MIM.
- The current method performs best when prototypes are larger than actual categories.
- No discussion on trade-offs between non-parametric and parametric methods.
- Experimental results show marginal improvements over previous baselines.

## Conclusions
Based on the reviews, the author feedback and a short postrebuttal discussion, the paper has been improved during the rebuttal period addressing some concerns of the reviewers and could be accepted in ICML if there is room in the program. Mainly, experiments  are very interesting showing interesting results although they are difficult to visualize the most outperforming. In addtion, there is no clear difference between equation 1 and 2, only theta deletion from one equation to the other. Therefore, the paper should included all the comments from the reviewers if the paper is finally accepted.